# Improved physical performance in obesity-resistant rats compared to obesity-prone rats: Effects of different diets and metabolic analysis

Daniel Sesana da Silva[1], Matheus Corteletti dos Santos[1], Lucas Furtado Domingos[2], Jóctan Pimentel Cordeiro[3], Kiany Miranda[1], Maria Gabriela Siqueira Tavares[1], Késsia Cristina Carvalho Santos[2], Ana Paula Lima-Leopoldo[2,3], André Soares Leopoldo[1,2]*

1 Postgraduate Program in Physiological Sciences, Center of Health Sciences, Federal University of Espírito Santo, Vitória, Espírito Santo, Brazil, 2 Postgraduate Program in Nutrition and Health, Center of Health Sciences, Federal University of Espírito Santo, Vitória, Espírito Santo, Brazil, 3 Center of Physical Education and Sports, Federal University of Espírito Santo, Vitória, Espírito Santo, Brazil

* andre.leopoldo@ufes.br

## Abstract

Animal models, such as high-fat diet-induced obese (DIO) rats, have been used to understand its pathophysiology. These models reveal differences between obesity-prone (OP) and obesity-resistant (OR) phenotypes. Thus, OR exhibit lower body mass gain and higher levels of physical activity, suggesting a more efficient energy metabolism. This study evaluated the metabolic adaptations and physical performance of OP and OR rats. Wistar rats (30 days old) were subjected to 23-week obesity exposure protocols. Initially, rats were randomized into two groups: a) SD: fed a standard diet (n = 39) and b) HFD: fed a high-fat diet (n = 39). Subsequently, animals were characterized as OP and OR on their respective diets: SD-OR (n = 13); SD-OP (n = 13); HFD-OP (n = 13); HFD-OR (n = 13). Nutritional, metabolic, and adiposity parameters were analyzed. Basal metabolism assessment was performed using indirect calorimetry. Physical performance and aerobic capacity were determined through treadmill exercise tests with gas analyzers for maximal oxygen consumption ($VO_2$). OR animals had lower body mass compared to OP animals, despite consuming the same caloric intake under both diets. HFD-OP rats gained 30.5% more weight than HFD-OR rats, while SD-OP rats gained 19.5% more than SD-OR rats. SD-OR rats gained 20.5% more weight than HFD-OR. No significant differences in adiposity indices were observed among groups. HFD-OR rats showed 15.6% higher $VO_2$max than HFD-OP rats; SD-OR rats had 12.8% higher $VO_2$max and 20.3% longer time to exhaustion compared to SD-OP rats. Indirect calorimetry revealed higher energy expenditure in OR animals during the dark cycle. OP animals exhibited elevated insulin and HOMA-IR levels, while OR animals had higher leptin and glucagon levels. In conclusion, OR rats showed improved physical performance and aerobic capacity compared to OP rats, even on a high-fat diet, suggesting that OR rats have adaptive mechanisms that enhance energy metabolism and endurance.

**Data availability statement:** All relevant data are within the manuscript and its Supporting Information files.

**Funding:** This study was supported by the Espírito Santo Research and Innovation Support Foundation – FAPES (grant numbers: 2022-5SBS2, 2022-J72BB and 2022-VT4KM) and the National Council for Scientific and Technological Development – CNPq.

**Competing interests:** The authors have declared that no competing interests exist.

## Introduction

Obesity is a complex metabolic condition characterized by excessive body fat accumulation resulting from interactions among genetic, behavioral, and environmental factors [1,2]. Obesity-resistant (OR) and obesity-prone (OP) phenotypes reflect differences in susceptibility to weight gain and the ability to maintain body weight under varying dietary conditions [3]. Previous studies have shown that OR individuals and animals have a greater ability to resist weight gain even when exposed to high-calorie diets, whereas OP individuals tend to accumulate more fat under the same conditions [4,5].

Previous studies suggest that OR is associated with greater metabolic efficiency and increased lipid oxidation, which may enhance physical performance [6,7]. Akieda-Asai et al. (2013) [5] found that in OR animals, the cross-sectional area (CSA) of adipocytes in white adipose tissue (WAT) is similar to that of control animals and smaller than that observed in OP animals. These OR animals also exhibited reduced expression of the enzyme fatty acid synthase (FAS) and increased expression of the protein carnitine palmitoyl transferase (CPT-1), indicating lower fat deposition and higher lipid oxidation [5]. During physical exercise, high lipid availability increases the utilization of fatty acids (FAs) for ATP production, which enhances muscle performance and reduces carbohydrate use, potentially decreasing fatigue [8,9]. Novak, Kotz & Levine (2006) [7] show that OR rats have a lower respiratory quotient (RQ), indicating greater efficiency in fatty acid metabolism as an energy source. Therefore, improved lipid oxidation efficiency may result in improved physical performance [9].

Moreover, studies on intrinsic aerobic capacity (IAC) models have demonstrated that animals with high aerobic capacity are resistant to weight gain (OR), whereas those with low aerobic capacity tend to be more susceptible to weight gain (OP) [10–12]. One possible explanation involves the redirection of pathways that optimize energy use by skeletal muscles during exercise, which may utilize internal processes more prevalent in OR individuals [13]. Additionally, studies suggest that adaptive thermogenesis varies among individuals, with animals of high aerobic capacity exhibiting greater adaptive thermogenesis than those of low aerobic capacity, which may contribute to the development of OR and OP phenotypes, respectively [10,12].

However, Milhem et al. (2024) [3] show that the metabolic flexibility of muscles to oxidize fat may compromise muscle function. OR animals exhibit higher spontaneous physical activity and oxygen consumption, but this does not necessarily imply superior physical performance [3]. High-fat diets can cause irreversible alterations in muscle function [3]. Further research is still needed to better understand how resistance and susceptibility to obesity affect energy metabolism, aerobic capacity, and physical performance, as well as to evaluate the benefits and limitations of these phenomena in obesity management and health.

Considering that, previous studies suggest better metabolic efficiency in OR rats, the aim of the present study was to investigate whether OR rats exhibit a more efficient basal metabolism and better physical performance due to a greater lipid

oxidation capacity, while OP rats display metabolic deficiencies that impair their physical capacity and increase suscepti-bility to obesity-related comorbidities.

Parte superior do formulário

Parte inferior do formulário

## Materials and methods

### Animal care

Male Wistar rats (n = 78; 30 days old) were individually housed in polypropylene cages with chrome-plated wire lids, lined with sterilized Pinus shavings. The animals were maintained under controlled environmental conditions, including a constant temperature of 24 ± 2°C, humidity of 55 ± 5%, and a reversed 12-hour light/dark cycle (11:00 PM to 11:00 AM). All experimental procedures followed the guidelines established by Law 11.794, October 8, 2008, Decree No. 6.899, July 15, 2009, as well as the norms of the National Council for Control of Animal Experimentation (CONCEA). The experimental protocol was approved by the Ethics Committee on Animal Use (CEUA) of the Federal University of Espírito Santo in a meeting held on October 1, 2021.

### Experimental protocol

After a period of 7 days of acclimatization, rats were initially randomized into two groups: 1) fed a standard diet (SD, n = 39) and; 2) fed a saturated high-fat diet (HFD, n = 39). SD group animals received a standard rodent diet containing 15.46% of calories from fat, 61.86% from carbohydrates, and 22.68% from proteins (Nuvilab CR1-Nuvital). HFD rats received a saturated high-fat diet containing 45.33% of calories from fat, 40.29% from carbohydrates, and 14.38% from proteins. The HFD was designed and formulated based on the AIN-93M standard diet with the addition of pork lard, following international standards for obesity induction [14]. These experimental diets provided sufficient amounts of proteins, vitamins, and minerals according to the Nutrient Requirements for Laboratory Animals [15]. All animals had free access to water and chow (40 g/day), with daily food consumption measured. Feed efficiency (FE) was calculated by dividing the total weight gain of the animals (g) by the total ingested energy (kcal) [16,17]. Caloric intake was calculated by multiplying daily food consumption by the caloric value of each diet (g x kcal). The experimental protocol spanned a total period of 23 weeks, divided into two phases, as demonstrated in Fig 1C: induction (3 weeks) and exposure to the diet and characterization of OR and OP (20 weeks). The onset of obesity (3 weeks) was determined based on previous studies conducted by our laboratory [18,19], considering the initiation of obesity when there was a significant increase in body weight among HFD rats compared to SD rats.

### Composition of experimental diets

The standard diet Nuvilab CR-1 consisted of the following ingredients: ground whole corn, soybean meal, wheat bran, calcium carbonate, dicalcium phosphate, sodium chloride, vitamin A, vitamin D3, vitamin E, vitamin K, vitamin B1, vitamin B2, vitamin B6, vitamin B12, niacin, calcium pantothenate, folic acid, choline chloride, iron sulfate, manganese sulfate, zinc sulfate, copper sulfate, calcium iodate, cobalt sulfate, lysine, methionine, and BHT. The high-fat diet was designed and formulated based on the AIN-93M standard diet with the addition of lard, following international standards for obesity induction [14]. Macronutrient percentages and caloric density of each experimental diet are presented in Supplement (S1 and S2 Tables).

Parte inferior do formulário

Parte superior do formulárioParte inferior do formulário

### Criteria for composition and redistribution of groups

At the initial stage of obesity (3 weeks), homogeneous groups were formed using a criterion based on body mass to redistribute the SD and HFD groups. The groups were redistributed into new groups and classified under OR and OP

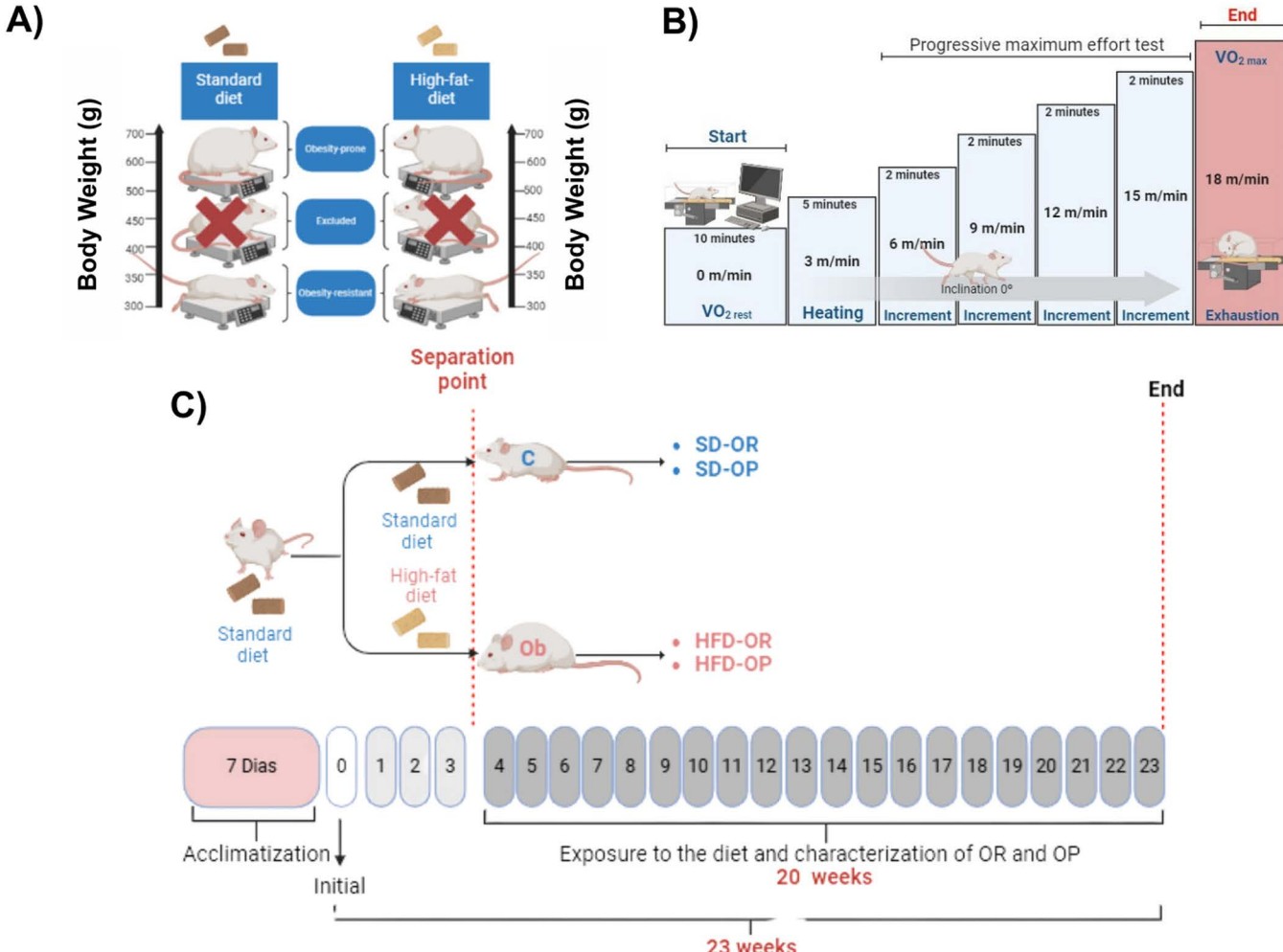

**Fig 1. Experimental design. (A)** Schematic representative of the group classification criterion by tercile based on animal body weight; **(B)** Progressive maximal effort test; **(C)** Experimental protocol.

conditions, irrespective of the experimental diet. Throughout the exposure period to the experimental diets and characterization of OR and OP (Fig 1A), the SD and HFD groups were redistributed based on tercile criteria, considering body mass. Thus, animals allocated to the upper tercile were classified as Obesity-prone (SD-OP and HFD-OP, respectively), while those in the lower tercile were characterized as Obesity Resistant (SD-OR and HFD-OR, respectively). Animals in the central tercile were excluded from the present study and utilized in other laboratory studies and/or additional experimental technique training (Fig 1A). The tercile classification method has been previously proposed to group and distinguish experimental groups [20–22]. Following the application of the aforementioned classification criterion, the SD and HFD groups were reallocated into four new groups: Standard Diet – Obesity-prone (SD-OP, n = 13); Standard Diet – Obesity-resistant (SD-OR, n = 13); High-Fat Diet – Obesity-prone (HFD-OP, n = 13); High-Fat Diet – Obesity-resistant (HFD-OR, n = 13). Animals remained in these conditions for an additional 20-week period termed as the diet exposure and OR and OP characterization phase (Fig 1C).

## Nutritional assessment

The nutritional profile was determined by analyzing body weight (BW), body fat (BF) and adiposity index (AI). The BW was measured weekly, and the amount of body fat was determined by the sum of epididymal, retroperitoneal and visceral (mesenteric fat) fat deposits. AI was calculated by the formula = total body fat/ final body weight, multiplied by 100 [23,24].

## Glycemic profile

In the twenty-third week of the experimental protocol, rats underwent a 6-hour fasting period. Blood samples from the tail artery were collected at baseline and after intraperitoneal administration of 25% glucose (Sigma-Aldrich, St. Louis, MO, USA), equivalent to 2g/kg. Blood samples were collected at time 0, considered as the baseline condition, and at 30, 60, 90, and 120 minutes after glucose administration. Glucose tolerance in these animals was evaluated by the area under the curve for glucose [25].

## Evaluation of Basal Metabolism and Physical Performance

To familiarize the animals with treadmill running, rats were subjected to adaptation sessions on a specific rodent treadmill (BONTHER – Ribeirão Preto, SP, Brazil) for five consecutive days, at a speed of 10 m/min for 10 minutes per day. All animals performed the $VO_{2max}$ test at least 48 hours after the final familiarization session. The assessment of maximal oxygen consumption ($VO_{2max}$) was conducted using a progressive treadmill protocol adapted from previous studies [26,27] during weeks 22 and 23 of the experimental protocol. Animals were first acclimated to the treadmill for 10 minutes before the beginning of the test, during which basal $VO_2$, basal $VCO_2$, and the respiratory quotient (RQ) were measured. Subsequently, the animals underwent a 5-minute warm-up at a low speed (3 m/min). The test started at a speed of 6 m/min, with increments of 3 m/min every 2 minutes (Fig 1B).

The criteria for determining $VO_{2max}$ included: stabilization of the oxygen consumption curve despite increasing workload, a respiratory exchange ratio greater than 1.05, or when the animal remained on the electric stimulus grid (1.5 µA) without resuming running within 15 seconds. $VO_{2max}$ was defined as the highest $VO_2$ value achieved during the test. Additionally, the maximum speed, total distance covered, and test duration were recorded to determine aerobic capacity and physical performance.

## Euthanasia

At the end of the experimental protocol (23 weeks), animals underwent a 6–8-hour fasting period and were subsequently anesthetized with ketamine hydrochloride (70 mg/kg/ip; Dopalen®, Sespo Indústria e Comércio Ltda – Vetbrands Division, Jacareí, São Paulo, Brazil) and xylazine hydrochloride (10mg/kg/ip; Anasedan®, Sespo Indústria e Comércio Ltda – Vetbrands Division, Jacareí, São Paulo, Brazil). Upon confirming absence of nociceptive reflexes after anesthesia induction, a lethal overdose (three times the induction dose) of ketamine and xylazine was administered (UNIFESP Euthanasia Guide, 2019). Following euthanasia, blood samples were collected in Falcon tubes, centrifuged at 5000 rpm for 10 minutes (Eppendorf Centrifuge 5804-R, Hamburg, Germany), and, the plasma was stored at −80°C (ColdLab Ultra Freezer CL374-86V, Piracicaba, São Paulo, Brazil).

## Biochemical and hormonal profile

To analyze the lipid and hormonal profiles, the animals were fasted for 6–8 hours. Serum concentrations of total cholesterol (TC), high-density lipoprotein (HDL) and low-density lipoprotein (LDL) were determined using specific kits (Bioclin Bioquímica®, Belo Horizonte, Minas Gerais, Brazil and Synermed do Brasil Ltda, São Paulo, Brazil). The results of the colorimetric enzymatic tests were determined using a BS200 automated biochemical analyzer (Mindray do Brasil – Comércio e Distribuição de Equipamentos Médicos Ltda, São Paulo, Brazil). Hormone concentrations of leptin, glucagon and insulin were determined by the ELISA method, using specific kits (Linco Research Inc, St. Louis, MO, USA), according to the

manufacturer's instructions. The readings were taken using a microplate reader (Spectra MAX 190, Molecular Devices, Sunnyvale, CA, USA).

## Histological Analysis of Visceral Adipose Tissue

Visceral adipose tissue samples were excised, weighed, and fixed in tubes containing 4% paraformaldehyde in phosphate-buffered saline (PBS, pH 7.0). The tissue fragments were dehydrated in graded ethanol solutions, cleared in xylene, and embedded in paraffin. Sections of 5 μm thickness were prepared and stained with hematoxylin-eosin (HE). Images were captured at 10×magnification using a microscope (Leica Mikroskopie & System GmbH, Wetzlar, Germany) equipped with a digital video camera, connected to a computer with image analysis software (Image Pro-plus, Media Cybernetics, Silver Spring, MD, USA). Image quantification was performed using ImageJ software (version 1.43u, National Institutes of Health, USA). Adipocyte area was calculated as the mean value obtained from measurements across 10 randomly selected fields per group [28]. For hyperplasia analysis, the mean adipocyte volume per field was determined [29,30].

## Statistical analysis

Data was displayed using descriptive measures of position and variability and subjected to analysis of variance (ANOVA) (one-way or two-way) for independent samples when appropriate. When significant differences were found (p<0.05), a Tukey *post hoc* test was carried out. The level of significance was 5%.

## Results

The evolution of animal body weight throughout the experimental protocol and during the exposure and characterization of OR and OP is shown in Fig 2. Groups SD and HFD exhibited similar body weights at the beginning of the experimental

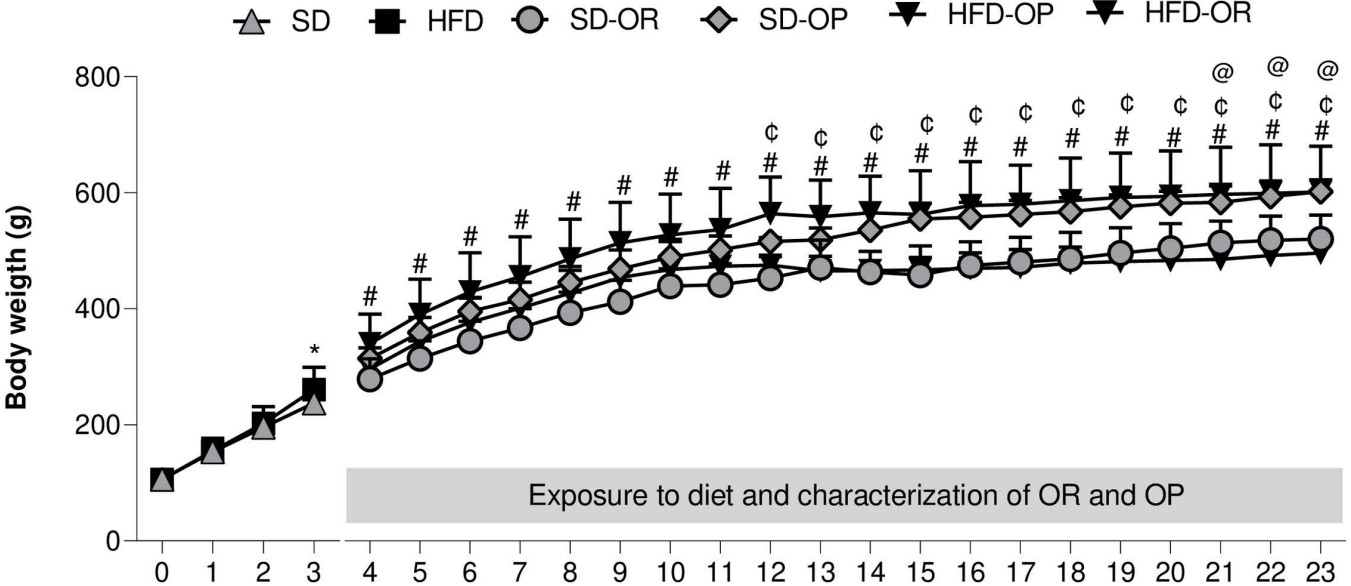

**Fig 2. Evolution of animal body weight throughout the entire experimental protocol.** Diet exposure and characterization of OR (Obesity Resistant) and OP (Obesity Prone) (Weeks 4-23). Standard diet (SD, n=39), high-fat diet (HFD, n=39), obesity-resistant standard diet (SD-OR, n=11), obesity-prone standard diet (SD-OP, n=13), obesity-resistant high-fat diet (HFD-OR, n=10), and obesity-prone high-fat diet (HFD-OP, n=11). Data are expressed as mean±standard deviation. * p<0.05 – SD *vs.* HFD; #SD-OR *vs.* SD-OP; ¢HFD-OR *vs.* HFD-OP; @SD-OR *vs.* HFD-OR. Two-way ANOVA for repeated measures, followed by Bonferroni *post-hoc* test.

protocol and maintained this similarity until the second week. However, starting from the third week (*defined as the onset of obesity*), the HFD group showed a significant increase in body weight compared to group SD (p < 0.05). Animals in the SD-OP group had higher body weight than those in the SD-OR group from the fourth week, which persisted until the 23rd week (*characterization of OR and OP*). The HFD-OP group showed significant differences in body weight from week 12 through week 23 (HFD-OP > HFD-OR), indicating that OP animals have higher body weight than OR animals regardless of the diet administered. Furthermore, during the last 3 weeks of the experimental protocol (21st to 23rd week), there was also a significant difference in OR condition according to the diet administered (SD-OR > HFD-OR).

Considering the evolution of body weight, our next step was to investigate the parameters of body weight, fat and adiposity of the experimental groups, as shown in Table 1. The Initial Body Weight (IBW) and Final Body Weight (FBW) data are consistent with the weekly body mass evolution, indicating that OP animals in the SD group had higher IBW (4th week) (+12.5%) and FBW (+15.7%) compared to OR animals. Similarly, HFD-OP animals showed higher IBW (+14.8%) and FBW (+21.1%) values compared to HFD-OR animals, respectively. No significant differences were observed in the between-group diet effects for IBW and FBW. Body weight gain was higher in the SD-OP group compared to the SD-OR group (+19.5%). Similarly, it was higher in the HFD-OP group compared to the HFD-OR group (+30.5%). Interestingly, the SD-OR group showed a higher gain in body weight compared to the HFD-OR group (+20.5%). Regarding body adiposity values, no significant differences were found between groups for epididymal, retroperitoneal, and visceral fat deposition, as well as total body fat and adiposity index.

To determine the functional significance of this increase in body weight, we measured the nutritional profile of the experimental groups. Animals on the SD had higher food consumption and caloric intake compared to those on the HFD in both the OR and OP conditions (Fig 3A and 3B). Moreover, SD-OP animals had higher food consumption and caloric intake than SD-OR animals. However, there was no statistically significant difference in feed efficiency, although higher values were observed in SD-OP (Fig 3C). In contrast, OP animals on HFD showed significantly higher values and a marked difference compared to OR animals on HFD, indicating that HFD-OP animals have a greater capacity to convert ingested energy into body weight (Fig 3C).

Fig 4 illustrates the morphometric analysis of visceral adipose tissue adipocytes. Animals from the HFD-OP group exhibited significantly larger adipocytes compared to the HFD-OR group (Fig 4A). No significant differences in adipocyte size were observed between the OR groups (SD-OR *vs.* HFD-OR) or between animals fed the standard diet (SD-OR *vs.* SD-OP). Furthermore, the number of adipocytes per field did not differ significantly among the groups (Fig 4C).

**Table 1. Characterization of Resistance to Obesity.**

| | Experimental Groups | | | |
| --- | --- | --- | --- | --- |
| | Standard Diet | | High-fat diet | |
| Variables | OR (n = 11) | OP (n = 13) | OR (n = 10) | OP (n = 11) |
| Initial Body Weight (g) | 279 ± 35 | 314 ± 17# | 296 ± 36 | 340 ± 51⁋ |
| Final Body Weight (g) | 520 ± 41 | 602 ± 21# | 496 ± 27 | 601 ± 79⁋ |
| Body Weight Gain (g) | 241 ± 31 | 288 ± 25# | 200 ± 26@ | 261 ± 42⁋ |
| Epididymal fat (g) | 9.39 ± 2.61 | 8.54 ± 1.03 | 9.32 ± 2.04 | 11.2 ± 4.2 |
| Retroperitoneal Fat (g) | 12.9 ± 4.1 | 11.9 ± 3.6 | 14.0 ± 4.0 | 17.5 ± 9.3 |
| Visceral fat (g) | 8.50 ± 3.79 | 8.20 ± 2.45 | 7.06 ± 1.60 | 9.58 ± 3.43 |
| Body fat (g) | 30.8 ± 9.9 | 28.7 ± 5.3 | 30.4 ± 6.0 | 38.3 ± 16.1 |
| Adiposity index (%) | 5.84 ± 1.56 | 4.77 ± 0.89 | 6.11 ± 1.12 | 6.21 ± 1.60 |

OR: obesity-resistant; OP: obesity-prone. Data expressed as mean ± standard deviation. p < 0.05 - #SD-OR vs. SD-OP; ⁋HFD-OR vs. HFD-OP; @SD-OR vs. HFD-OR. Two-way ANOVA, supplemented with Tukey's *post-hoc* test.

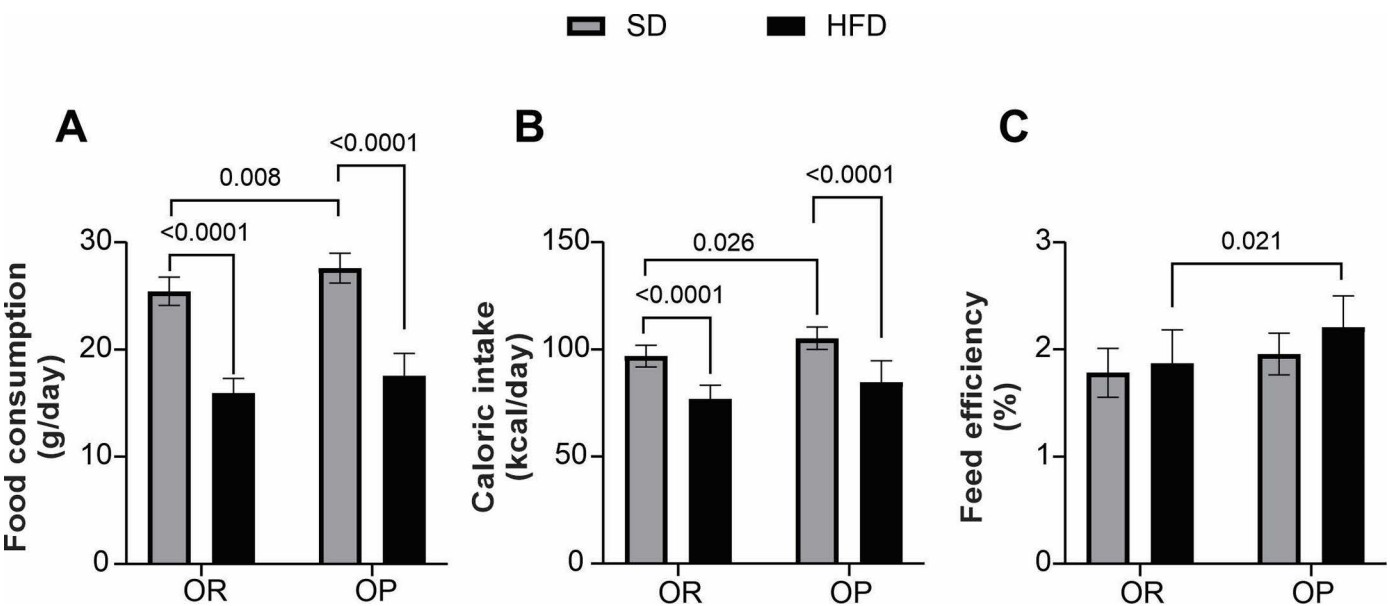

**Fig 3. Nutritional profile during the characterization period of OR and OP. (A)** Average daily food consumption; **(B)** Average daily caloric intake; **(C)** Feed efficiency (%). Standard diet obesity-resistant group (SD-OR, n = 11), standard diet obesity-prone group (SD-OP, n = 13), hyperlipidic diet obesity-resistant group (HFD-OR, n = 10), and hyperlipidic diet obesity-prone group (HFD-OP, n = 11). Data expressed as mean ± standard deviation. Two-way ANOVA, supplemented with Tukey's *post-hoc* test.

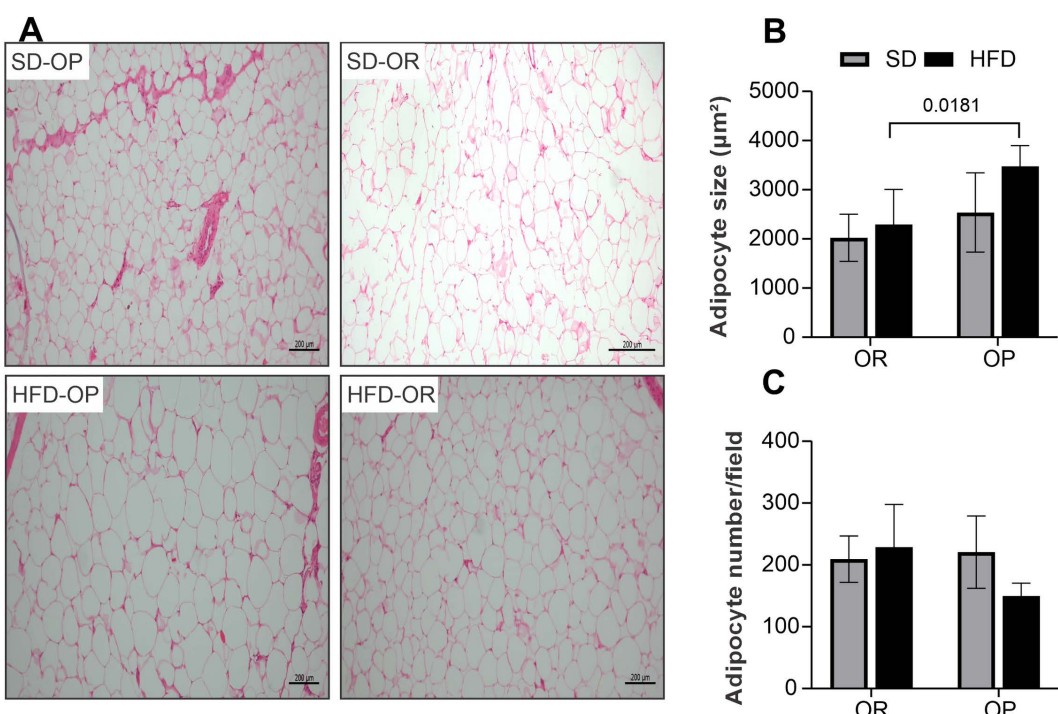

**Fig 4. Histological analysis and quantification of adipocyte morphology in visceral adipose tissue. (A)** Representative hematoxylin-eosin (HE) stained sections of visceral adipose tissue; **(B)** Quantification of adipocyte size (μm²); **(C)** Quantification of adipocyte number per field. Standard diet obesity-resistant group (SD-OR, n = 6), standard diet obesity-prone group (SD-OP, n = 6), hyperlipidic diet obesity-resistant group (HFD-OR, n = 6), and hyperlipidic diet obesity-prone group (HFD-OP, n = 6). Data expressed as mean ± standard deviation. Two-way ANOVA, supplemented with Tukey's *post-hoc* test.

To assess causality between obesity and metabolic disorders, we examined glycemic and insulinemic profiles shown in Fig 5. Significant between-group differences were observed in the GTT at 30 minutes after-glucose overload administration (Fig 5A) between the SD-OR and HFD-OR groups. No significant differences between groups were found at

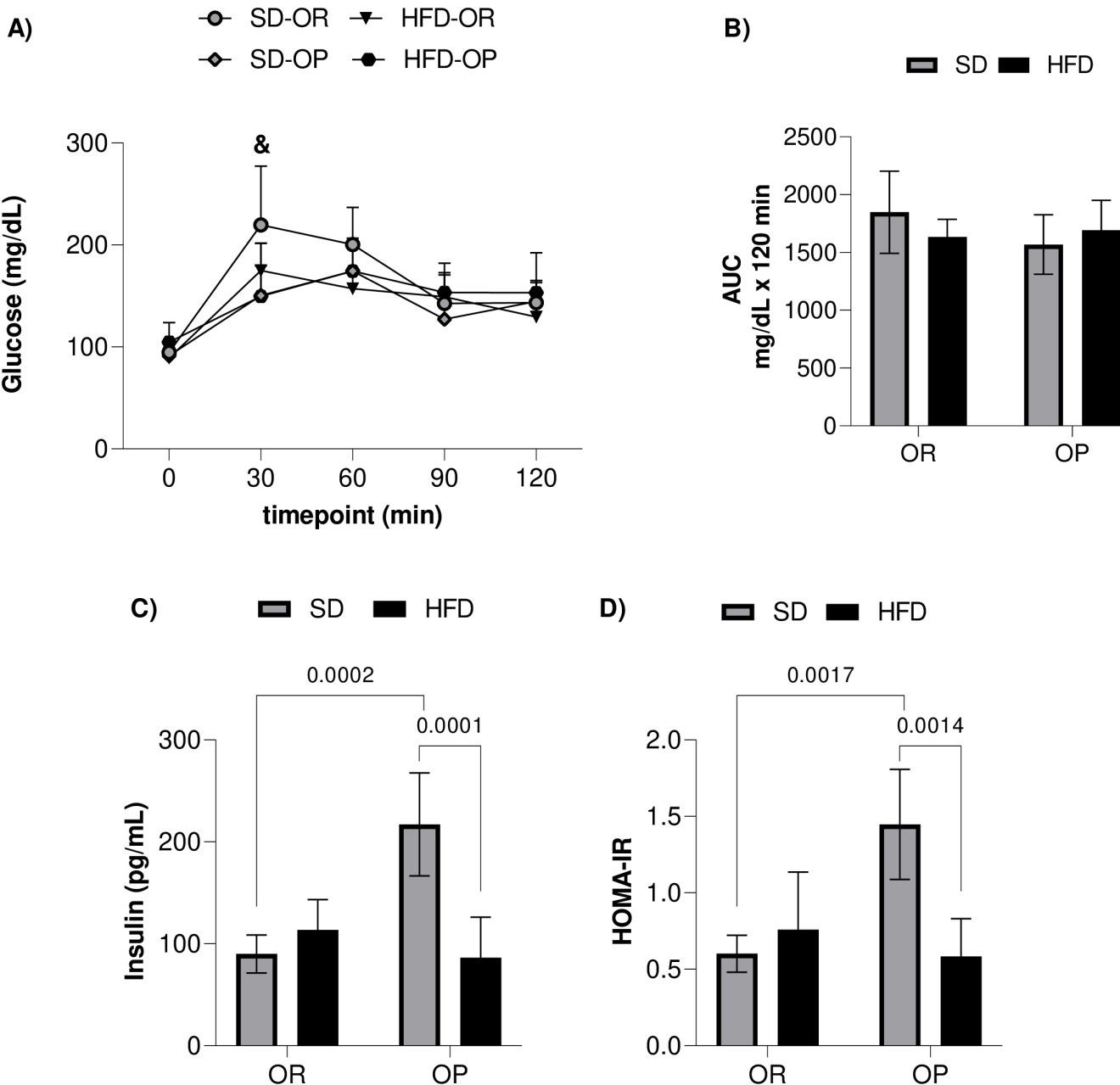

**Fig 5. Glycemic and insulinemic profiles of the experimental groups. (A)** Glucose Tolerance Test (GTT); **(B)** Glycemic Area Under the Curve (AUC). Standard diet obesity-resistant group (SD-OR, n = 10), standard diet obesity-prone group (SD-OP, n = 11), hyperlipidic diet obesity-resistant group (HFD-OR, n = 10), and hyperlipidic diet obesity-prone group (HFD-OP, n = 8). **(C)** Fasting serum insulin; **(D)** Homeostasis Model Assessment – Insulin Resistance (HOMA-IR). Standard diet obesity-resistant group (SD-OR, n = 5), standard diet obesity-prone group (SD-OP, n = 5), hyperlipidic diet obesity-resistant group (HFD-OR, n = 6), and hyperlipidic diet obesity-prone group (HFD-OP, n = 5). Data expressed as mean ± standard deviation. $p < 0.05$ - &SD-OR vs. HFD-OR. Two-way repeated measures ANOVA, supplemented with Bonferroni's post-hoc test (GTT). Two-way ANOVA, supplemented with Tukey's post-hoc test (AUC; Insulin and HOMA-IR).

other time points during the test, nor in the AUC (Fig 5B). In the insulinemic profile, the SD-OP group had elevated fasting serum insulin levels compared to the SD-OR (+141%) and HFD-OP (+151.3%) groups, respectively (Fig 5C). In addition, the SD-OP group also had higher HOMA-IR levels than the SD-OR (+140.4%) and HFD-OP (+147.4%) groups (Fig 5D).

Next, we investigated the biochemical and hormonal profile of the experimental groups. As shown in Table 2, SD-OR animals had higher (+32.9%) total cholesterol levels compared to HFD-OR animals. In addition, SD-OP animals showed increased HDL levels (+22.9%) compared to HFD-OP animals. However, no significant differences in LDL levels were observed between the groups. The SD-OR group had significantly higher leptin levels (+163.3%) compared to the SD-OP group. Similarly, the HFD-OP group also showed a higher level (+139.7%) of this parameter compared to SD-OP. Regarding glucagon levels, the results showed that the SD-OR group had significantly higher levels compared to the HFD-OR (+77.7%) and SD-OP (+62.8%) groups, respectively.

To investigate the relationship between maximal oxygen consumption (VO$_2$max), physical performance, and phenotypic differences among the experimental groups, we conducted specific metabolic and functional assessments, as illustrated in Fig 6. Initially, we analyzed basal metabolic parameters, including basal oxygen consumption (VO$_2$basal), basal carbon dioxide production (VCO$_2$basal), and basal respiratory quotient (RQ) (Fig 6A–C). However, no significant differences were observed between groups under basal conditions for VO$_2$ and VCO$_2$. (Fig 6A and B). The results also showed that HFD-OP animals exhibited a lower RQ compared to SD-OP animals, indicating a predominance of fat utilization as an energy substrate (Fig 6C). Nevertheless, no significant differences in RQ were found among the other groups. Additionally, the HFD-OP group showed higher VO$_2$ compared to the SD-OP group (Fig 6D). Moreover, OR animals subjected to the HFD displayed greater final running speed and total distance covered compared to OP animals on the same diet (Fig 6E and 6F). Finally, the SD-OR group demonstrated a longer relative test time compared to the SD-OP group (Fig 6G), and similarly, the HFD-OR group also exhibited a longer relative test time compared to the HFD-OP group.

## Discussion

The aim of this study was to investigate the metabolic characteristics and physical performance under conditions of obesity-resistant (OR) and obesity-prone (OP) phenotypes, independent of the diet administered. In addition, the study aimed to evaluate whether OR rats have a more efficient basal metabolism due to their increased lipid oxidation capacity, which may improve their physical performance. Conversely, the study sought to determine whether OP rats have metabolic impairments that could adversely affect their physical performance and increase their susceptibility to obesity-related comorbidities.

In this context, the literature suggests that OP and OR phenotypes relate to how individuals respond to the development or prevention of obesity, particularly when exposed to calorie-dense diets. Within a given environment, significant

**Table 2. Biochemical and hormonal profile.**

| | Experimental Groups | | | |
|---|---|---|---|---|
| | Standard Diet | | High-fat diet | |
| Variables | OR (n = 11) | OP (n = 12) | OR (n = 10) | OP (n = 10) |
| Cholesterol (md/dL) | 59.8 ± 11.8 | 47.5 ± 12.1 | 45.0 ± 16.3@ | 42.3 ± 9.6 |
| HDL (md/dL) | 18.9 ± 2.6 | 19.8 ± 3.4 | 16.1 ± 1.7 | 16.1 ± 4.6£ |
| LDL (md/dL) | 8.4 ± 2.7 | 9.3 ± 2.8 | 12.9 ± 5.7 | 13.0 ± 6.1 |
| Leptin (ng/mL)¥ | 2.45 ± 0.32 | 0.93 ± 0.25# | 2.60 ± 0.36 | 2.23 ± 0.63£ |
| Glucagon (ng/mL)¥ | 0.15 ± 0.05 | 0.10 ± 0.01# | 0.09 ± 0.02@ | 0.09 ± 0.03 |

OR: obesity-resistant; OP: obesity-prone. HDL: high-density lipoprotein; LDL: low-density lipoprotein.

¥SD-OR (n = 5); SD-OP (n = 5); HFD-OR (n = 6); HFD-OP (n = 6). Data expressed as mean ± standard deviation. p < 0.05 - @SD-OR *vs.* HFD-OR; #SD-OR *vs.* SD-OP; £SD-OP *vs.* HFD-OP. Two-way ANOVA, followed by Tukey's *post-hoc* test.

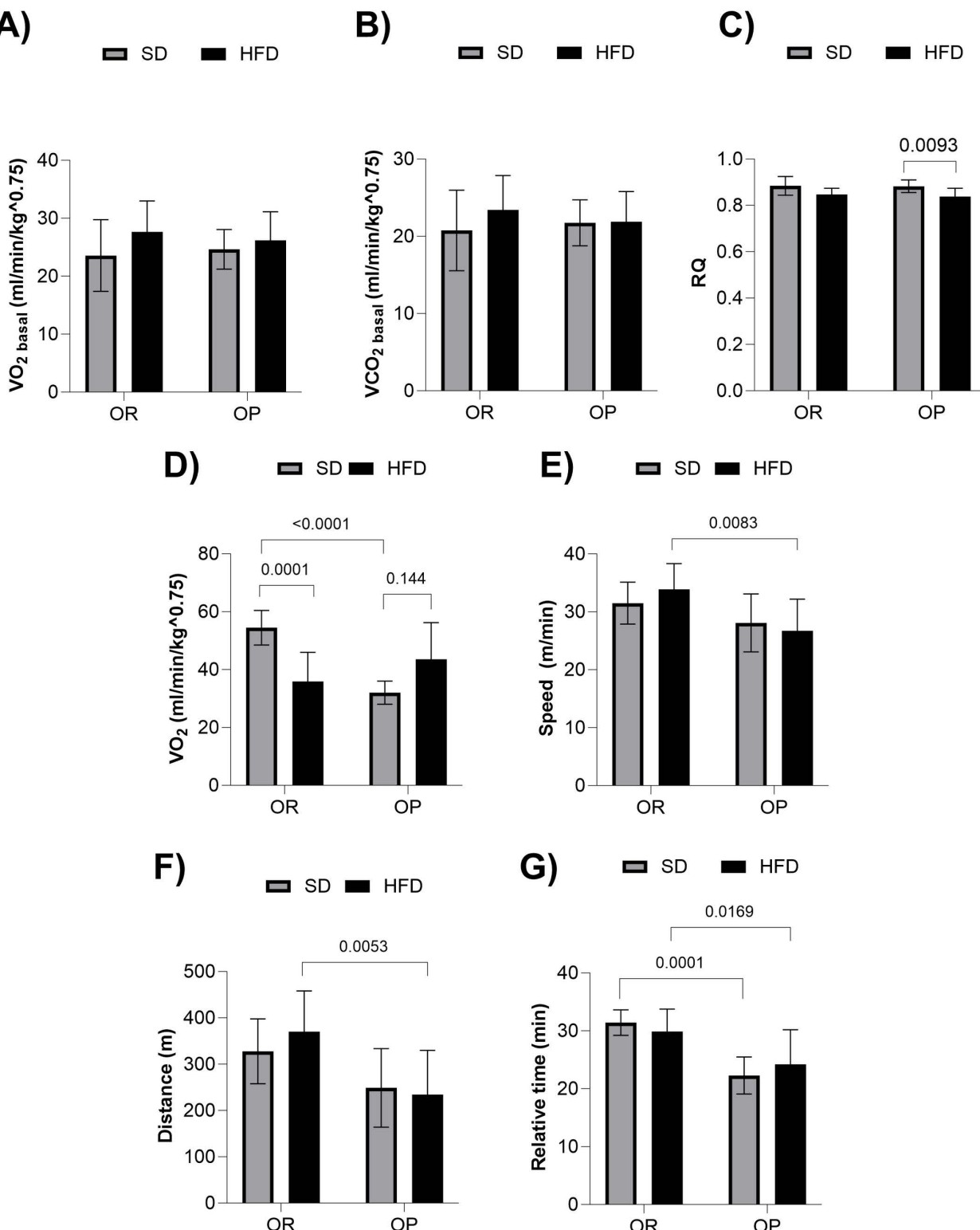

**Fig 6. Evaluation of basal metabolic parameters and physical performance in experimental groups. (A)** Basal oxygen consumption ($VO_{2basal}$); **(B)** Basal carbon dioxide production ($VCO_{2basal}$); **(C)** Basal respiratory quotient (RQ); **(D)** Maximal oxygen volume ($VO_{2max}$); **(E)** Total speed of the physical performance test (m/min); **(F)** Total distance covered in the physical performance test (m); **(G)** Total duration of the physical performance test (min);

– Groups: standard diet obesity-resistant (SD-OR, n = 11), standard diet obesity-prone (SD-OP, n = 12), high-fat diet obesity-resistant (HFD-OR, n = 9), and high-fat diet obesity-prone (HFD-OP, n = 11). Data are expressed as mean ± standard deviation. P < 0.05 – * Significant difference between the 1st and 2nd test, within the group; #SD-OR *vs.* SD-OP; @SD-OR *vs.* HFD-OR; £SD-OP *vs.* HFD-OP. Two-way ANOVA, followed by Tukey's *post-hoc* test.

differences in body weight and fat between individuals suggest that adiposity is shaped by interactions among various metabolic, behavioral, and environmental factors [3]. In summary OR is characterized by the ability to attenuate body weight gain and fat deposition [31], even when consuming a high-fat-diet. Conversely, obesity susceptibility refers to an individual's tendency to develop excess body weight and/or obesity in response to extrinsic factors such as high caloric intake, poor dietary habits, and/or a sedentary lifestyle [4,32].

Interestingly, in the present study, animals classified as OP showed greater weight gain than the OR group, without significant changes in overall body composition and, therefore, did not develop obesity according to the standard definition established by the World Health Organization, which characterizes obesity as an abnormal or excessive accumulation of body fat that poses a health risk [33]. In this context, OP animals had higher total body weight and increased visceral adipose tissue cross-sectional area compared to OR animals; however, they did not differ significantly in adiposity index, as expected. Thus, in the context of this study, the terms OP and OR refer to relative differences in body weight gain and visceral fat accumulation, rather than the classic definition of obesity based on an overall increase in body fat. Nevertheless, the differences observed in visceral fat accumulation and body weight, together with other relevant metabolic and functional results, support the scientific relevance of the experimental model and the reported results.

In obesity research, rodents, especially those subjected to a high-fat diet (HFD), are widely used due to their efficacy in developing an obese phenotype [34]. In the current study, animals were randomly assigned and started the experimental protocol with equivalent body weight. However, differences became apparent following the increase in caloric density of the HFD (4.83 *vs.* 3.88 kcal/g). Over a 3-week period, significant increases in body weight were observed in animals fed the HFD, particularly after the third week of exposure. These results demonstrate that the HFD used in this study was effective in inducing obesity, consistent with previous research from our [9,19,35] and other studies in the field [18,36]. Milhem et al. (2024) [3] reported similar results in C57BL/6J mice, with a 59.4% increase in body weight compared to controls (standard diet). This increase included an 11.5% increase in lean body weight and a 56.8% increase in fat mass due to the high fat content of the experimental diet.

The results from this study show that OP animals experienced greater increases in body weight compared to OR animals, regardless of whether they were on a standard diet (SD) or an HFD. Specifically, OP animals on the SD showed a 12.5% increase in initial body weight (IBW) and a 15.7% increase in final body weight (FBW). In contrast, OP animals on the HFD showed an increase of 14.8% in IBW and 21.1% in FBW compared to OR animals (Table 2). Despite the higher food intake and caloric consumption observed in the SD group, there were no significant differences in energy efficiency between the groups (Fig 3).

The results of this study reveal an interesting and partially divergent finding from the existing literature, as hyperphagia was not observed in animals exposed to the high-fat diet (HFD). Food consumption and caloric intake were higher in the groups fed a standard diet (SD) compared to those fed an HFD throughout the experimental period (Fig 2A and 2B). However, feeding with an HFD resulted in similar energy efficiency despite lower caloric intake, demonstrating a greater conversion of ingested energy into body weight gain. This phenomenon can be explained by differences in nutrient-induced thermogenesis, favoring fat deposition in HFD-fed animals [37,38]. In contrast, SD-fed animals require substantially higher food consumption to achieve comparable energy efficiency.

The rapid rise in obesity prevalence can be partially attributed to the widespread availability of highly palatable, energy-dense foods, leading to food intake that exceeds energy requirements. However, previous studies have shown that high-fat diets, when not combined with other components such as sucrose or reinforcing flavors, may be less palatable to rodents, thereby reducing spontaneous intake [39,40].

Contrary to our initial hypothesis, the results obtained were unexpected. Although OP animals exhibited greater body weight gain compared to OR animals under their respective diets, no significant differences were observed in adipose tissue depots or adiposity index between OP and OR groups (Table 1). These findings contrast with previous results reported by our research group [24,35]. However, we observed that HFD-OP animals presented a greater visceral adipose tissue area compared to HFD-OR animals (Fig 4A), despite similar caloric intake between groups. These findings highlight the importance of adipose tissue, particularly visceral fat, in regulating energy homeostasis. Under chronic exposure to an HFD, adipose tissue expands through both hypertrophy and hyperplasia [41]. Adipocyte hypertrophy is directly associated with increased inflammation and the development of insulin resistance [42,43].

In agreement with our findings, Akieda-Asai et al. (2013) [5] observed that OR animals exhibit a transverse adipocyte sectional area similar to control groups, but significantly smaller than that of OP animals. Additionally, the same authors demonstrated that OR animals show lower expression of fatty acid synthase (FAS) and elevated levels of carnitine palmitoyltransferase-1 (CPT-1), suggesting that the OR phenotype is associated with decreased lipogenesis and enhanced lipid oxidation, respectively. However, our findings partially diverge from these observations, as no significant differences in the respiratory quotient (RQ) were detected between OP and OR groups. Moreover, the evaluation of OR animals should consider not only the lower body weight gain and adiposity but also the maintenance of caloric intake comparable to that of OP animals.

Supporting this evidence, Poret et al. (2018) [43] reported that OP rats exhibited greater adipocyte cross-sectional area (CSA) and larger visceral fat depots compared to OR rats. Furthermore, HFD feeding was shown to induce increased expression of pro-inflammatory cytokines in OP animals. These findings reinforce the hypothesis that OP rats have a higher predisposition to developing comorbidities, a condition that may be further exacerbated by exposure to a high-fat diet.

Several studies have examined discrepancies in energy in relation to factors such as resting metabolic rate (RMR), the thermic effect of food (TEF), and physical activity [44]. Some researchers have emphasized a central relationship between energy expenditure and physical activity [1,7,10,11,45,46], while others have focused on target cells such as skeletal muscle and adipose tissue [5,6]. In addition, spontaneous physical activity and oxygen consumption may influence the energy expenditure in OR rats [3,22]. Recent studies have also suggested a relationship between metabolism and the microbiome [3]. Interestingly, we did not observe differences in resting $VO_2$ between groups, contrary to our initial hypothesis that OR animals would have higher resting $VO_2$ compared to OP animals, as suggested by previous studies [3,22].

Intrinsic capacities for lipid storage (lipogenesis) and oxidation (lipolysis) are influenced by a combination of hormonal factors, including leptin, insulin, and glucagon, which regulate energy metabolism and mitochondrial function. In addition, gut microbiota permeability and hypothalamic responses that control appetite and physical activity play critical roles. Although these capacities are interrelated, making the processes of fat gain or loss more complex than a simple energy balance, they directly influence metabolic efficiency and the tendency to accumulate fat [4,41,47–50].

Contrary to our initial hypothesis, no significant differences were observed in glucose tolerance profiles between experimental groups, including the OP animals fed a HFD, which were expected to have impaired glucose tolerance. This finding is consistent with recent studies from our laboratory that also showed no glucose intolerance in SD groups [9]. Nevertheless, OP animals on HFD had higher blood glucose levels compared to OP animals on SD [36]. High levels of fatty acids can increase insulin secretion in response to glucose, although prolonged exposure can inhibit this secretion in vitro [51]. Short-term high-fat feeding induces insulin resistance but not hyperglycemia in rats [52]. Therefore, while excess fatty acids and glucose stimulate insulin release to convert glucose into glycogen and fat, the presence of ectopic fat in obese individuals contributes to peripheral insulin resistance.

Interestingly, SD-OP animals exhibited hyperinsulinemia and insulin resistance, with higher fasting serum insulin and HOMA-IR levels compared to both SD-OR and HFD-OP groups. This suggests that the high carbohydrate content (>60%) of the SD likely contributed to increased triglycerides and insulin resistance, consistent with the literature indicating that high-carbohydrate diets increase fasting insulin and resistance [53,54].

Our study showed that leptin significantly influences energy metabolism and metabolic health. In disagreement with our initial hypothesis, we did not observe glucose intolerance in OP animals on an HFD. Although the SD-OP group had lower leptin levels, there were no significant differences in fat deposition between groups, consistent with the observation that leptin levels correlate with adipose tissue in both humans and rodents [55,56]. Reduced leptin production is associated with a lower basal metabolic rate, which makes weight loss more difficult and increases the risk of conditions such as type 2 diabetes and insulin Resistance [57].

Conversely, SD-OP animals exhibited hyperinsulinemia and insulin resistance, with elevated serum insulin and HOMA-IR levels, suggesting that a high-carbohydrate diet exacerbates insulin resistance and promotes obesity [53,54]. The reduced leptin levels in the SD-OP group, along with reduced aerobic performance, indicate an increased risk of metabolic disease and a reduced life expectancy [58,59]. In addition, leptin resistance may be present from birth, manifesting as reduced leptin levels and predispose animals to increased sensitivity, that is not solely induced by a high-calorie diet [1,55]. Overall, both low leptin production and leptin resistance have significant implications for metabolic health, influencing susceptibility to weight gain and related diseases.

The data presented in Fig 6 provide important insights into $VO_2$ and physical performance between experimental groups. The results show that OR (obesity-resistant) rats had significantly higher $VO_2$ compared to both OP (obesity-prone) rats and HFD-OR (obesity-resistant on a high-fat diet) rats. This finding suggests that the OR condition is associated with improved aerobic capacity, which is consistent with studies showing a distinct metabolic profile in obesity-resistant rats, often showing higher $VO_2$ [3,11].

Furthermore, HFD-OP rats had higher $VO_2$ than SD-OP rats, which may reflect a metabolic adaptation to a high-fat diet. This finding is consistent with literature suggesting that while the OR condition is associated with increased metabolic capacity, adaptation to a high-fat diet can also increase $VO_2$, without necessarily indicating greater resistance to obesity [9].

In terms of physical performance, HFD-OR rats showed greater final speed and total distance covered compared to HFD-OP rats. This suggests that the OR condition, even under a high-fat diet, is associated with higher physical performance [5]. This improvement may be due to metabolic and physiological adaptations that allow obesity-resistant animals to better cope with the challenges of a high-fat diet.

Additionally, the SD-OR group showed a longer relative time in physical activities than the SD-OP group, and the HFD-OR group showed a longer relative time than the HFD-OP group. These results suggest that the OR condition may be associated with an increased ability to sustain prolonged physical activity, supporting the literature linking OR to increased physical capacity [7,22].

## Conclusion

In conclusion, this study demonstrates that obesity-resistant (OR) rats exhibit improved physical performance and greater aerobic capacity than obesity-prone (OP) rats, even when exposed to high-fat diet. These findings highlight the complex interaction between genetic predisposition, metabolic efficiency, and physical performance, and suggest that OR rats possess adaptive mechanisms that confer advantages in energy metabolism and physical endurance. Further research is needed to elucidate the precise mechanisms underlying these adaptations and their implications for obesity management and metabolic health.

## Supporting information

**S1 Table. Composition of macronutrients in the standard diet and high-fat diet.** *Standard diet for rodents. BHT – Butylated Hydroxytoluene.
(PDF)

**S2 Table. Macronutrient percentages and caloric density.** *Standard diet for rodents. Parte superior do formulário. (PDF)

## Acknowledgments

We express our gratitude to Technician Raissa Corrêa Andrade for her valuable contribution to the completion of this project and collaborations with LUCCAR/UFES and LHT/UFES for their support in conducting histomorphological analysis.

## Author contributions

**Conceptualization:** Daniel Sesana da Silva, Matheus Corteletti dos Santos, Ana Paula Lima-Leopoldo, André Soares Leopoldo.

**Data curation:** Daniel Sesana da Silva, Matheus Corteletti dos Santos, Lucas Furtado Domingos, Jóctan Pimentel Cordeiro, Kiany Miranda, Maria Gabriela Siqueira Tavares, Késsia Cristina Carvalho Santos, Ana Paula Lima-Leopoldo.

**Formal analysis:** Daniel Sesana da Silva, Matheus Corteletti dos Santos, Lucas Furtado Domingos, Jóctan Pimentel Cordeiro, Kiany Miranda, Maria Gabriela Siqueira Tavares, Késsia Cristina Carvalho Santos, Ana Paula Lima-Leopoldo, André Soares Leopoldo.

**Funding acquisition:** André Soares Leopoldo.

**Investigation:** Daniel Sesana da Silva, Matheus Corteletti dos Santos, Lucas Furtado Domingos, Jóctan Pimentel Cordeiro, Kiany Miranda, Maria Gabriela Siqueira Tavares, Késsia Cristina Carvalho Santos, Ana Paula Lima-Leopoldo, André Soares Leopoldo.

**Methodology:** Daniel Sesana da Silva, Matheus Corteletti dos Santos, Lucas Furtado Domingos, Jóctan Pimentel Cordeiro, Kiany Miranda, Maria Gabriela Siqueira Tavares, Késsia Cristina Carvalho Santos, Ana Paula Lima-Leopoldo, André Soares Leopoldo.

**Project administration:** Ana Paula Lima-Leopoldo, André Soares Leopoldo.

**Resources:** Daniel Sesana da Silva, Matheus Corteletti dos Santos, Lucas Furtado Domingos, Jóctan Pimentel Cordeiro, Kiany Miranda, André Soares Leopoldo.

**Validation:** Maria Gabriela Siqueira Tavares, Késsia Cristina Carvalho Santos, Ana Paula Lima-Leopoldo.

**Visualization:** Daniel Sesana da Silva, Matheus Corteletti dos Santos, Jóctan Pimentel Cordeiro, Ana Paula Lima-Leopoldo, André Soares Leopoldo.

**Writing – original draft:** Daniel Sesana da Silva, Matheus Corteletti dos Santos, Lucas Furtado Domingos, Jóctan Pimentel Cordeiro, Kiany Miranda, Maria Gabriela Siqueira Tavares, Késsia Cristina Carvalho Santos, Ana Paula Lima-Leopoldo, André Soares Leopoldo.

**Writing – review & editing:** Daniel Sesana da Silva, Matheus Corteletti dos Santos, Lucas Furtado Domingos, Jóctan Pimentel Cordeiro, Kiany Miranda, Maria Gabriela Siqueira Tavares, Késsia Cristina Carvalho Santos, Ana Paula Lima-Leopoldo, André Soares Leopoldo.

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
