## [Decision Letter · Decision Letter 0]

PONE-D-24-57591Improved Physical Performance in Obesity-Resistant Rats Compared to Obesity-Prone Rats: Effects of Different Diets and Metabolic AnalysisPLOS ONE

Dear Dr. Soares Leopoldo,

Thank you for submitting your manuscript to PLOS ONE. After careful consideration, we feel that it has merit but does not fully meet PLOS ONE’s publication criteria as it currently stands. Therefore, we invite you to submit a revised version of the manuscript that addresses the points raised during the review process.

We look forward to receiving your revised manuscript.

Kind regards,

Dr. V. V. Sathibabu Uddandrao

Academic Editor

PLOS ONE

Reviewers' comments:

Reviewer's Responses to Questions

**Comments to the Author**

1. Is the manuscript technically sound, and do the data support the conclusions?

Reviewer #1: No

Reviewer #2: Yes

2. Has the statistical analysis been performed appropriately and rigorously? 

Reviewer #1: Yes

Reviewer #2: No

3. Have the authors made all data underlying the findings in their manuscript fully available?

Reviewer #1: Yes

Reviewer #2: Yes

4. Is the manuscript presented in an intelligible fashion and written in standard English?

Reviewer #1: Yes

Reviewer #2: Yes

5. Review Comments to the Author

Reviewer #1: The study has a major problem. The aim was comparison of obesity-prone (OP) vs. obesity-resistant (OR) rats, and any interpretation and discussion builds on the assumption that this was done. However, the employed protocol did not deliver groups of OP and OR rats. Furthermore, the HFD exerted very different action vs. HFD used in other studies, which hampers interpretation in the light of previous studies.

*) It appears that this study did NOT compare obesity-prone vs. obesity-resistant rats as the authors claim. Obesity is a state of excessive fat accumulation, but the “adiposity index” indicative of relative fat mass is not different between the OP and OR rats. Hence, lean and fat mass are increased proportionally in the “OP” vs. the “OR” groups examined in this study, which means that the authors have actually compared rats that grew bigger vs. smaller, without any difference in their proneness to develop obesity.

*) The effects of the HFD formulation used in this study had very different effects on Wistar rats vs. HFD used in previous studies. Albeit HFD caused a (marginal) increase in body weight vs. SD after the initial 3 weeks, this was abolished after 23 weeks, which clearly differs from the effects of HFD used in other studies, which has been described to cause a marked and persistent increase in the body weight of Wistar rats. The reason for this difference is unclear (low palatability of HFD used here?). Interpretation and discussion in the paper do not address this matter, but are rather based on the assumption that HFD used here is the same as HFD used by others.

*) It may be good to use a different term for “adiposity index”, since adiposity index is an established term for an index based on hip circumference in humans which may cause confusion.

*) Were all the lipid and hormone parameters measured from blood collected during killing? It is stated that rats were 8h fasted for lipid analysis, but no such information is given for the hormone measurements. Also consider that glycemia and, hence, also insulinemia are severely deranged in response to anesthesia.

*) Figs. 1, 2, and 3 showing the protocol details could be fused into Fig.1A,B,C. This might be clearer for the reader.

*) The term “Killing” might be more appropriate than “Euthanasia” (which means killing to prevent suffering).

*) In line 512 it is speculated that resistance to obesity may be associated with increased lipid oxidation. This does not fit with the lack of a difference in RQ between the OP and OR rats.

*) Fig. 7C: Under the given circumstances, the RQ should be between 0.7 and 1.0, but the depicted values are far above this range. Metabolic cages are sometimes not 100% precise in this regard, but the numbers shown here are very far from the appropriate range.

*) It appears that graph C and D have been swapped in the legend to Fig.8.

*) Figure are very poorly pixelated, almost unreadable.

Reviewer #2: In the current study, the authors evaluated that the “Improved Physical Performance in Obesity-Resistant Rats Compared to Obesity-Prone Rats: Effects of Different Diets and Metabolic Analysis.” overall manuscript is well written,

6. PLOS authors have the option to publish the peer review history of their article (what does this mean? ). If published, this will include your full peer review and any attached files.

**Do you want your identity to be public for this peer review?** For information about this choice, including consent withdrawal, please see our Privacy Policy .

Reviewer #1: No

Reviewer #2: **Yes: ** Brahmanaidu Parim

---

## [Author Response · Author response to Decision Letter 1]

5 May 2025

The responses to the Reviewers have been attached in a file.

---

## [Decision Letter · Decision Letter 1]

PONE-D-24-57591R1Improved Physical Performance in Obesity-Resistant Rats Compared to Obesity-Prone Rats: Effects of Different Diets and Metabolic AnalysisPLOS ONE

Dear Dr. Soares Leopoldo,

Thank you for submitting your manuscript to PLOS ONE. After careful consideration, we feel that it has merit but does not fully meet PLOS ONE’s publication criteria as it currently stands. Therefore, we invite you to submit a revised version of the manuscript that addresses the points raised during the review process.

We look forward to receiving your revised manuscript.

Kind regards,

V. V. Sathibabu Uddandrao

Academic Editor

PLOS ONE

Journal Requirements:

Additional Editor Comments :

While the reviewer notes that the study may not represent high-level science, most of the issues raised in the initial review have been addressed satisfactorily. However, the reviewer cannot support acceptance as long as the authors address the reviewer's concern.

Reviewers' comments:

Reviewer's Responses to Questions

**Comments to the Author**

1. If the authors have adequately addressed your comments raised in a previous round of review and you feel that this manuscript is now acceptable for publication, you may indicate that here to bypass the “Comments to the Author” section, enter your conflict of interest statement in the “Confidential to Editor” section, and submit your "Accept" recommendation.

Reviewer #1: (No Response)

Reviewer #2: All comments have been addressed

2. Is the manuscript technically sound, and do the data support the conclusions?

Reviewer #1: Partly

Reviewer #2: Yes

3. Has the statistical analysis been performed appropriately and rigorously? 

Reviewer #1: Yes

Reviewer #2: Yes

4. Have the authors made all data underlying the findings in their manuscript fully available?

Reviewer #1: Yes

Reviewer #2: Yes

5. Is the manuscript presented in an intelligible fashion and written in standard English?

Reviewer #1: Yes

Reviewer #2: Yes

6. Review Comments to the Author

Reviewer #1: The paper has improved in many details, but there is one point that I still regard as essential:

Meaningful communication builds on agreement about the terms we use. Maybe a small number of authors in the past have failed to use the term “obese” appropriately, but this is opposed by thousands of publications, which adhered to the definition found in every serious scientific source, which unmistakably defines the term “obesity” is as a state of excessive body fat (not of increased weight with normal body composition; see WHO, NIH, Mayo, etc.). If you insist to use “obese” in a sense that differs from the generally agreed definition, I would at least claim a straightforward statement pinpointing this deviation. This could, e.g., be a sentence that reads as follows or similar: “Note that our rats referred to as obesity-prone grew bigger than the obesity-resistant without a change in body composition and, hence, did not develop obesity according to the usual definition of an increased relative fat mass.”

And the sentence in lines 185,186 seems incomplete: "The adiposity 185 index was calculated by dividing total body fat by final body mass and multiplying the result by ??? (24,25)."

Reviewer #2: As per the instructions, all suggestions were incorporated in the revised manuscript "Improved Physical Performance in Obesity-Resistant Rats Compared to Obesity-Prone Rats: Effects of Different Diets and Metabolic Analysis"

7. PLOS authors have the option to publish the peer review history of their article (what does this mean? ). If published, this will include your full peer review and any attached files.

**Do you want your identity to be public for this peer review?** For information about this choice, including consent withdrawal, please see our Privacy Policy .

Reviewer #1: No

Reviewer #2: No

---

## [Author Response · Author response to Decision Letter 2]

12 Jun 2025

The responses to the reviewers were attached as a file.

---

## [Decision Letter · Decision Letter 2]

Improved Physical Performance in Obesity-Resistant Rats Compared to Obesity-Prone Rats: Effects of Different Diets and Metabolic Analysis

PONE-D-24-57591R2

Dear Dr. André Soares Leopoldo,

We’re pleased to inform you that your manuscript has been judged scientifically suitable for publication and will be formally accepted for publication once it meets all outstanding technical requirements.

Kind regards,

V. V. Sathibabu Uddandrao

Academic Editor

PLOS ONE

Additional Editor Comments (optional):

Reviewers' comments:

Reviewer's Responses to Questions

**Comments to the Author**

1. If the authors have adequately addressed your comments raised in a previous round of review and you feel that this manuscript is now acceptable for publication, you may indicate that here to bypass the “Comments to the Author” section, enter your conflict of interest statement in the “Confidential to Editor” section, and submit your "Accept" recommendation.

Reviewer #1: All comments have been addressed

2. Is the manuscript technically sound, and do the data support the conclusions?

Reviewer #1: Yes

3. Has the statistical analysis been performed appropriately and rigorously? 

Reviewer #1: Yes

4. Have the authors made all data underlying the findings in their manuscript fully available?

Reviewer #1: Yes

5. Is the manuscript presented in an intelligible fashion and written in standard English?

Reviewer #1: Yes

6. Review Comments to the Author

Reviewer #1: (No Response)

7. PLOS authors have the option to publish the peer review history of their article (what does this mean? ). If published, this will include your full peer review and any attached files.

**Do you want your identity to be public for this peer review?** For information about this choice, including consent withdrawal, please see our Privacy Policy .

Reviewer #1: No

---

## [Editor Report · Acceptance letter]

PONE-D-24-57591R2

PLOS ONE

Dear Dr. Soares Leopoldo,

I'm pleased to inform you that your manuscript has been deemed suitable for publication in PLOS ONE. Congratulations! Your manuscript is now being handed over to our production team.

Kind regards,

on behalf of

Dr. V. V. Sathibabu Uddandrao

Academic Editor

PLOS ONE